# Design and Characterizations of Inhalable Poly(lactic-*co*-glycolic acid) Microspheres Prepared by the Fine Droplet Drying Process for a Sustained Effect of Salmon Calcitonin

**DOI:** 10.3390/molecules25061311

**Published:** 2020-03-13

**Authors:** Hideyuki Sato, Aiko Tabata, Tatsuru Moritani, Tadahiko Morinaga, Takahiro Mizumoto, Yoshiki Seto, Satomi Onoue

**Affiliations:** 1Laboratory of Biopharmacy, School of Pharmaceutical Sciences, University of Shizuoka, 52-1 Yada, Suruga-ku, Shizuoka 422-8526, Japan; h.sato@u-shizuoka-ken.ac.jp (H.S.); m12074@u-shizuoka-ken.ac.jp (A.T.); seto@u-shizuoka-ken.ac.jp (Y.S.); 2Functional Material Development Center, Imaging Engine Development Division, RICOH Company, Ltd., 16-1 Honda-machi, Numazu-shi, Shizuoka 410-1458, Japan; tatsuru.moritani@jp.ricoh.com (T.M.); tadahiko.morinaga@jp.ricoh.com (T.M.); 3ILS Inc., 1-2-1 Kubogaoka, Moriya, Ibaraki 302-0104, Japan; takahiro.mizumoto@ils.co.jp

**Keywords:** dry powder inhaler, fine droplet drying process, poly(lactic-*co*-glycolic acid), salmon calcitonin, sustained release

## Abstract

The present study aimed to develop inhalable poly (lactic-*co*-glycolic acid) (PLGA)-based microparticles of salmon calcitonin (sCT) for sustained pharmacological action by the fine droplet drying (FDD) process, a novel powderization technique employing printing technologies. PLGA was selected as a biodegradable carrier polymer for sustained-release particles of sCT (sCT/SR), and physicochemical characterizations of sCT/SR were conducted. To estimate the in vivo efficacy of the sCT/SR respirable powder (sCT/SR-RP), plasma calcium levels were measured after intratracheal administration in rats. The particle size of sCT/SR was 3.6 µm, and the SPAN factor, one of the parameters to present the uniformity of particle size distribution, was calculated to be 0.65. In the evaluation of the conformational structure of sCT, no significant changes were observed in sCT/SR even after the FDD process. The drug release from sCT/SR showed a biphasic pattern with an initial burst and slow diffusion in simulated lung fluid. sCT/SR-RP showed fine inhalation performance, as evidenced by a fine particle fraction value of 28% in the cascade impactor analysis. After the insufflation of sCT samples (40 µg-sCT/kg) in rats, sCT/SR-RP could enhance and prolong the hypocalcemic action of sCT possibly due to the sustained release and pulmonary absorption of sCT. From these observations, the strategic application of the FDD process could be efficacious to provide PLGA-based inhalable formulations of sCT, as well as other therapeutic peptides, to enhance their biopharmaceutical potentials.

## 1. Introduction

Recently, a number of biologics including peptides, proteins, antibodies, and nucleic acids, have been developed as pharmaceutical candidates for new modalities [1], since they can act on targeted molecules that cannot be controlled by small molecule drugs in providing greater safety, potency, and target specificity [2,3]. For the administration of the biologics, parenteral routes like intravenous and/or intramuscular injections have been mainly used due to their poor oral bioavailability and stability in the gastrointestinal tract, which can adversely affect patient compliance. As an alternative non-invasive route for the administration of biologics, pulmonary delivery has recently attracted interests from pharmaceutical researchers [1,4,5,6]. Pulmonary drug delivery has not only been used for topical effects within the lung tissues but also for systemic therapeutic actions because of the characteristics of the respiratory tract, including a large surface area, abundant capillary network, thin membrane with adequate permeability for macromolecules, reduced enzymatic degradation, and a lack of first pass metabolism [7,8]. Among the many types of formulations for pulmonary administration, dry powder inhaler (DPI) is one of the promising dosage forms, since it can be more economical, environmentally friendly, and easy for use by patients themselves than the other forms [9,10,11]. To achieve desirable drug delivery by DPI systems to disease sites in the respiratory tract and/or absorption sites in the lung, controlling the powder properties, especially aerodynamic particle size and particle size distribution, is essential [12,13]. Generally, a size range of particles from 1 µm to 5 µm is considered to be appropriate to deposit in the deep lung for drug absorption from alveolar sites [14]. Despite these attractive aspects of pulmonary delivery for biologics, it is still challenging to strictly control the aerodynamic properties of drug-loaded powders at a single micron scale by conventional powderization technology to achieve efficient pulmonary delivery of drugs.

Recently, several ink jet printing technologies have been applied to the production of pharmaceutical formulations in the form of particles, hydrogels and films for different administration routes, including oral, transdermal, respiratory or parenteral routes [15,16,17], since the industrial inkjet printing has now reached high standards of flexible, robust, and reliable performance. The fine droplet drying (FDD) process, a new powderization technique employing an ink jet head used in the printing industry, was developed by our group and firstly applied to the production of pharmaceutical formulations [18]. In this process, the inkjet head utilizes a piezo element as an actuator to make uniform fine droplets, resulting in extremely uniform particles after drying the generated droplets [19]. The size of produced particles can be precisely controlled at a single-micron scale by changing the size of nozzle holes. Previously, this process was strategically applied to produce an amorphous solid dispersion of cyclosporine A (CsA), a typical biopharmaceutical classification system class II compound, leading to the successful development of a formulation with improved biopharmaceutical properties [18]. The CsA particles prepared with the FDD process had better powder properties than the reference formulation prepared by spray-drying, possibly due to the different mechanisms for producing fine droplets. In the FDD process, the resonance phenomenon of a sonic wave has been proposed as the driving force to produce droplets [19], contributing to the efficient production of powder products by a high resonant frequency (>300 kHz). Furthermore, the inkjet head is a scalable atomizer, possibly leading to high throughput production of powders. Although these characteristics of FDD process would be desirable for particle design and control aerodynamic properties of DPIs, far less is known about the applicability of the FDD process to particle design for inhalation. This study is the first attempt to apply the FDD process to design inhalable particles for pulmonary delivery of therapeutic peptides.

In this study, salmon calcitonin (sCT) was selected as a model peptide, consisting of 32-amino acids, with a short half-life in the body [20,21]. Poly (lactic-*co*-glycolic) acid (PLGA) with an average molecular weight of 12 kDa (Resomer^®^ RG502H) was used as a carrier polymer for preparing sustained-release particles of sCT (sCT/SR) with a prolonged systemic biological action. An FDD process using RICOH MH2420, an inkjet head, was conducted for the preparation of uniform fine particles. The prepared formulation was characterized in terms of physicochemical and in vitro inhalation properties. To evaluate in vivo efficacy and prolongation of the pharmacological action, the hypocalcemic effect of sCT was measured after the insufflation of sCT samples (40 µg-sCT/kg) in rats.

## 2. Results and Discussion

### 2.1. Preparation of sCT/SR by the FDD Process

For most of the therapeutic peptides, a short half-life in the systemic circulation is one of the key challenges in their clinical applications [2,21,22]. In the present study, to extend the duration of the pharmacological action of sCT, PLGA was selected as a biodegradable sustained-release carrier for preparing sCT/SR. For the steady production of sCT/SR, the temperature of air flow was set at 25 °C to avoid the aggregation of PLGA microparticles during the drying process because the glass transition point of PLGA is relatively low (around 40 °C), possibly resulting in improved yield of PLGA microspheres [23]. The loading efficiency of sCT into PLGA microsphere was 99.4% as assayed by HPLC analysis, indicating almost no degradation of sCT during the FDD process. The denaturation and/or degradation of the peptide drugs would be one of the significant concerns during the manufacturing process for preparing products due to their poor resistance to heat stress [24]. Regarding this point, owing to the mild drying conditions of the FDD process, the chemical degradation and denaturation of sCT was negligible even after the drying process. Based on SEM observation, the appearance of sCT/SR was spherical micron-sized particles without significant aggregation of the particles, and the median particle size was 3.6 µm measured by laser diffraction analysis (Figure 1). The SPAN factor, one of the parameters to present the uniformity of particle size distribution, was 0.6, indicating the uniform size distribution of sCT/SR prepared with the FDD process. In a previously study, an amorphous solid dispersion of CsA was developed by the FDD process, and the median diameter and SPAN factor of the particles were very similar (D_50_: 3.6 µm, SPAN: 0.4) [18]. Even using different kinds of excipients and solvent in the production, the FDD system could reproducibly generate micron-sized particles, because the size of the nozzle hole is thought to be one of the significant factors determining the particle size. The diameter of the nozzle hole on the surface of the inkjet head for the FDD process was 8 µm; thus, most particles would be of a single-micron size and show very uniform distribution after drying the fine droplets. As a DPI formulation, particle size and its uniformity are considered to be critical to characterize the inhalation performance. Thus, to control the particle size of DPI powders at a single-micron scale with a uniform size distribution, application of the FDD process might be an attractive way to design fine powders with fine inhalation properties. Although the ultrasonic nozzle has been used as a similar technique to produce uniform fine droplets through vibration of the nozzle tip by piezoelectric transducers, there are still some problems including low throughput and less control of the particle shape in this process [25]. On the other hand, for generating uniform fine droplets from the surface of the inkjet head in the FDD process, a high-frequency device with 310 kHz was employed to achieve highly efficient powder production.

The conformational structure of therapeutic peptides can influence their therapeutic potential because the interactions between the peptide and target molecules may vary depending on the structure, possibly leading to a reduction of the pharmacological action and even safety problems caused by peptide aggregates [26,27,28]. To evaluate the conformational changes in the secondary structure of sCT, circular dichroism (CD) analysis of sCT samples were conducted (Figure 2A). As shown in the CD spectrum, non-aged sCT solution showed a negative peak with a minimum at 202 nm, indicating that the protein assumed a random coil conformation in the solution, as reported previously [29]. Even after the FDD process, there were no significant differences between non-aged sCT solution and sCT/SR; thus, the changes in the structure of sCT were minimal during the process of preparing sCT/SR. sCT has been reported as a peptide with amyloidogenic capacity that is time-, concentration-, temperature- and solvent-dependent [30]. This property could cause the formation of insoluble fibrils during the manufacturing process and/or storage periods, and contribute to the reduction of the therapeutic potential and even cytotoxicity [31]. With respect to quality control and safety, such fibril formation during the manufacturing process should be avoided in the development of sCT-loaded DPI systems. Thioflavin T (ThT) is known as a specific fluorescence probe for amyloid-like aggregates due to its selective binding capacity against the sheet rich structure [32], and the ThT binding assay can be used to estimate the risk of aggregation caused by conformational changes into a β-sheet rich structure [26]. As the results from the ThT binding assay, aged sCT solutions showed higher ThT binding than fresh sCT solution, and the increases were concentration-dependent. Under the solution state, there could be a risk of forming the fibril structure of sCT. On the other hand, the differences of ThT binding between non-aged sCT solution and sCT/SR were not significant. Considering these points, the FDD process could be used to prepare sCT-loaded PLGA particles with no significant conformational changes and aggregation of sCT, possibly due to the mild conditions for drying the generated fine droplets.

### 2.2. Release Behavior of sCT from PLGA Particles in SLF

PLGA formulations for the sustained-delivery of peptides have been widely investigated and utilized to overcome the necessity of frequent administrations of peptides owing to poor pharmacokinetic profiles of macromolecular therapies [33,34]. In this study, the PLGA-based particles for therapeutic peptides were designed by the FDD process to extend the therapeutic action by prolonging the pulmonary absorption of sCT. To estimate the sustained-release property of sCT/SR in the respiratory tract, the release test in simulated lung fluid (SLF) was carried out (Figure 3). Due to the water-soluble property of sCT, sCT powder showed a very quick dissolution, as evidenced by almost 100% dissolution of sCT even at 15 min. The quick dissolution of sCT might cause excessive absorption after the administration of sCT formulations, possibly leading to unexpected side-effects caused by a marked increase in the blood concentration of sCT or short duration of systemic exposure to the drug due to its short half-life in the systemic circulation [35]. On the contrary, sCT/SR exhibited two-phase dissolution: an initial burst phase for the first 30 min and a sustained-release phase of sCT released from the matrix structure of PLGA-based microparticles. The first initial burst dissolution of sCT/SR might be related to the drug molecules adsorbed onto the surface of PLGA particles. PLGA has been used in many approved drug products to control the pharmacokinetic and pharmacodynamic behavior of target drugs because PLGA shows no major safety concerns owing to its biocompatible properties [3,36,37]. The mechanisms of drug release from PLGA particles are known depending on drug diffusion and matrix erosion of PLGA during the dissolution process. Although the SLF contains a component with solubilizing capacity like dipalmitoylphosphatidylcholine (DPPC) owing to its surfactant action, sustained release of sCT from the PLGA particles was observed in sCT/SR. Generally, the particle size and shape could influence the release behavior of included drugs depending on the variation in surface area of the particles [38,39]. With respect to this point, the FDD process can precisely control the particle size at a micron scale with a spherical shape, possibly contributing to the particle design, to achieve desirable release behavior depending on the clinical demands. For the peptide-loaded inhalable powder, this sustained-release system could contribute to enhancement of the duration of action.

### 2.3. Inhalation Properties of sCT/SR-Respirable Powder (sCT/SR-RP)

To achieve adequate absorption of drugs from the respiratory tract via DPI systems, the inhalation performance of an inhalable formulation is considered as a significant factor for efficient drug delivery to the absorption site in the lung [40]. Micron-sized particles like sCT/SR generally possess the potential to easily aggregate due to relatively high surface energy, possibly leading to impaired inhalation performance because of aggregation. To avoid such aggregation and improve flowability of sCT/SR, the carrier particles of lactose with a mean diameter of ca. 50 µm were mixed, and microparticles of sCT/SR were adhered on to the surface of a lactose carrier. The micron-sized particles can be adhered by van der Waals force and electrostatic interactions, and the shape and surface roughness of the particle are significant factors to determine the strength of adhesion force [41]. In the SEM image of sCT/SR-RP (Figure 4A), the aggregation of sCT/SR was negligible. Improved flow ability might contribute to the fine emission property of sCT/SR-RP from DPI capsules and enhanced dispersibility. To estimate the in vivo deposition of sCT after the inhalation of sCT/SR-RP, the in vitro inhalation properties were evaluated by Andersen cascade impactor analysis. The deposition pattern of sCT/SR-RP after inhalation with JetHelar^®^, a simple inhalation device, is shown in Figure 4B. The calculated aerodynamic diameter and fine particle fraction (FPF) value of sCT/SR-RP were 4.1 µm and 28%, respectively. It has been reported that particles with a range of aerodynamic diameter from 1 to 5 µm are likely to reach the deep lung after inhalation [14]; thus, sCT/SR-RP was considered a suitable aerodynamic particle size for the efficient pulmonary delivery of sCT/SR through the inhalation process. The FPF value is defined as the proportion of the total particle amount comprising particles with a diameter smaller than 5.0 µm, which can reach the deep lung under physiological inhalation conditions. Despite the sticky property of PLGA, the main excipient of sCT/SR, sCT/SR-RP demonstrated a fine inhalation performance as evidenced by calculated a FPF value of 28%. The spherical shape with a uniform size distribution of FDD particles might promote the fine dispersibility of DPI systems. Although particles with an irregular shape have been reported as having high adhesion properties in some reports [42], the particle shape is one of the most challenging factors to control in powder technology. Regarding this point, the FDD process could reproducibly produce the spherical particles with a narrow size distribution [18,19]. In addition to the powder properties such as particle diameter, shape, density and flowability, the inhalation device is also a key determinant of the dispersibility of DPI. For further improvement of the inhalation performance, selecting a suitable device might lead to higher delivery efficiency than shown by the present data. Based on these findings, sCT/SR-RP exhibited sustained-release and fine inhalation properties, possibly leading to appropriate pulmonary delivery of sCT by inhalation.

### 2.4. Hypocalcemic Action after Intratracheal Administration of sCT/SR-RP

For the treatment of osteoporosis and hypercalcemia, injections of sCT have been widely used because of poor absorption from the gastrointestinal tract and low chemical stability under gastric conditions due to the polypeptide nature [43]. Even after the parenteral administration of sCT, systemic exposure is very limited due to fast renal clearance as well as enzymatic degradation in the systemic circulation, necessitating frequent injections for adequate treatment [44]. The in vivo efficacies of sCT samples after intratracheal administration in rats (40 µg-sCT/kg, i.t.) were assessed in terms of the hypocalcemic action, the main pharmacological action of sCT (Figure 5). After the insufflation of control-RP composed of lactose, there were no significant changes in the plasma calcium concentration. In the sCT-RP-treated group, a temporary decrease of plasma calcium levels was observed immediately after intratracheal administration. However, the plasma calcium level increased up to almost 100% of the initial state at 4 h after the insufflation of sCT-RP, since the inhaled sCT might be completely dissolved and absorbed immediately after the administration, and almost eliminated after 4 h. On the contrary, sCT/SR-RP could extend the duration of the action even to 24 h after the administration due to its sustained-release property in the respiratory site corresponding to the result of dissolution testing. With respect to the intensity of the hypocalcemic effect, sCT/SR-RP exhibited a stronger action than sCT-RP, possibly owing to the better inhalation performance of sCT/SR-RP than that of sCT-RP. In addition to control of the release behavior, PLGA microspheres might also improve the chemical stability of encapsulated sCT in the respiratory tract by protecting from the outer environment, leading to the better pharmacological action than sCT-RP. The area under effective curve (AUEC) value of the hypocalcemic action for 24 h was calculated (Figure 5B). There was a slight reduction of the AUEC_0–24 h_ value in the sCT-RP-treated group compared with that in the control-RP group, but sCT/SR-RP could significantly decrease the AUEC_0–24 h_ value. This enhancement of the pharmacological action would be attributable to both the better inhalation performance of FDD particles and sustained-release behavior derived from the properties of PLGA. This PLGA-based DPI system could prolong the pharmacological actions of therapeutic peptides and consequently reduce the frequency of administration, and promote self-medication by patients themselves.

## 3. Materials and Methods

### 3.1. Chemicals

PLGA (Resomer^®^ RG-502H, Mw: 7000–17,000, Lactide:Glycolide = 50:50) was obtained from Evonik Industries AG (Essen, German), and sCT was kindly gifted from ILS Inc. (Ibaraki, Japan). Respitose^®^ SV-003, the lactose carrier particles for the DPI system, was supplied by DMV (Veghel, The Netherlands). All other reagents were purchased from commercial sources.

### 3.2. Preparation of sCT-loaded PLGA Micro Particles

The sCT-loaded PLGA particles were prepared in accordance with a previously reported process using the FDD process with some modifications [18]. Briefly, sCT (50 mg) and PLGA (Resomer^®^ RG-502H; 4950 mg) were dissolved in ethyl acetate (245 g; 2% *w/w* as solid contents). The solution was stirred for 1 h at a stirring speed of 1000 rpm before filtration through a 1-µm filter to remove insoluble materials. The filtrate was atomized with RICOH MH2420 (RICOH, Tokyo, Japan), passed through an inkjet head, and generated droplets were dried with air at 25 °C. The detailed conditions were as follows: air flow, 50 m^3^/h; temperature, 30 °C; nozzle hole diameter, 8 µm; and piezo frequency, 310 kHz.

The amount of sCT in each sample was assayed by the HPLC system with a UV detector (Shimadzu Co., Ltd., Kyoto, Japan). An Inertsil ODS-3 column (2 µm, 2.1 mm × 50 mm; GL science) was used at 40 °C. Acetonitrile (A) and milli Q containing 0.1% trifluoroacetic acid (B) were chosen as mobile phases for the analysis of sCT. The conditions of the mobile phase were as follows: flow rate, 0.25 mL/min; and isocratic condition, A:B = 50:50. A wavelength of 290 nm was used for the assay.

### 3.3. Preparation of Respirable Powders (RP) of sCT Samples

sCT/SR-RP was prepared by simple mixing with sCT/SR and Respitose^®^ SV-003, a lactose carrier with a mean diameter of 50 µm [45]. sCT/SR particles were gently mixed by pestle and mortar at a ratio of sCT/SR to the lactose carrier of 1:5 (*w/w*). For preparing sCT-RP, the reference formulation, a physical mixture of sCT and lactose (1:99 (*w/w*)), was micronized by pestle and mortar, followed by mixing with lactose carrier (1:5 (*w/w*)). Control-RP was also prepared by similar procedures except for using lactose instead of sCT.

### 3.4. Physicochemical Characterizations of sCT/SR

#### 3.4.1. Scanning Electron Microscopy

The surface morphology of sCT/SR was assessed using the scanning electron microscopy (TM3030, Hitachi Co., Ltd., Tokyo, Japan). The prepared powders were fixed onto the surface of an aluminum holder by carbon tape, and MSP-1S (Vacuum Device, Ibaraki, Japan), an ion sputtering device, was used for coating by Pt on the surface of sCT samples.

#### 3.4.2. Laser Diffraction

The particle size distribution of sCT/SR was determined using Microtrac MT3000II (MicrotracBel, Osaka, Japan) with a dry disperser. To estimate the uniformity of the distribution, the SPAN factor was calculated with the following equation: SPAN = (*d*_90_ − *d*_10_)/*d*_50_; *d*_10_, *d*_50_ and *d*_90_ are the particle diameters at 10%, 50% and 90% of the cumulative volume, respectively.

#### 3.4.3. Circular Dichroism (CD) Analysis

To check the conformational structure of sCT, CD analysis was conducted using the Jasco model J-820 (Jasco Co., Tokyo, Japan). Briefly, sCT samples were dissolved at 0.04 mg-sCT/mL in 50% MeOH/20 mM sodium phosphate buffer. The sample solutions were analyzed under the following conditions: path-length of quartz cell, 10 mm; spectral step resolution; 0.1 nm; scan speed; 50 nm/min.

#### 3.4.4. Thioflavin T (ThT) Binding Assay

Aggregation properties of sCT were evaluated fluorometrically by ThT binding [32]. As the positive control of aggregation, aged sCT solutions for 24 h at 37 °C were prepared with different concentrations (1, 5 and 10 mg/mL). In brief, sCT samples were dissolved in ethyl acetate, to which 0.01 M HCl was added to precipitate PLGA. After centrifuging at 15,000× *g* for 5 min, each supernatant was diluted by 50% MeOH/20 mM PBS (sCT concentration: 1 mg/mL). The 20 μL of each solution was added to 1980 μL of 5 μM ThT in 20 mM PBS (pH 6.0). Fluorescence intensity was immediately measured by an RF-1500 spectrofluorophotometer (Shimadzu, Kyoto, Japan) with λ_ex_ and λ_em_ of 450 and 482 nm, respectively.

#### 3.4.5. Dissolution Test of sCT in Simulated Lung Fluid (SLF)

To estimate the release behavior of sCT from PLGA particles in the respiratory system, a dissolution test of sCT samples was carried out using biorelevant media mimicking lung conditions. The composition of SLF was based on a previous report with some modifications [46]. The dissolution test was carried out in 10 mL of SLF with constant shaking at 30 spm using an Incubator M-100 (TAITEC Co., Nagoya, Japan) at 37 °C. The samples were weighed to give 1 mg of sCT in each sample tube. Each sample (100 μL) was collected at determined times (0.25, 0.5, 1, 2, 4, 6, 12 and 24 h), and centrifuged at 10,000 rpm for 5 min. After the centrifugation, each supernatant was diluted with the same volume of methanol. The concentration of sCT was measured by the HPLC/UV system, as described in Section 3.2.

### 3.5. Andersen Cascade Impactor Analysis

To estimate the inhalation properties of sCT/SR-RP, cascade impactor analysis was carried out using an AN-200 system (Shibata Scientific Technology, Tokyo, Japan) in accordance with USP 29 < 601 > AEROSOLS. Briefly, 30 mg of sCT/SR-RP was used to fill a JP No. 2 hard capsule of hydroxypropyl methylcellulose, and the capsule was installed in a JetHaler^®^ (Hitachi Unisia, Kanagawa, Japan). The formulation in each capsule was dispersed through the device at 28.3 L/min for 10 s × 3 times, and the amount of sCT in each stage (stages 0–7) and the capsule was measured using the HPLC system according to Section 3.2. The fine particle fraction (FPF) value was defined as the ratio of total drug deposited in stage 2 and lower.

### 3.6. Hypocalcemic Action of sCT Samples in Rats

#### 3.6.1. Animals

Male SD rats (180 ± 20 g, 6–7 weeks of age; Japan SLC, Shizuoka, Japan) were housed at 3 rats/cage with free access to food/water on a 12-h dark/light cycle with controlled temperature and humidity (24 ± 1 °C, 55% ± 5% RH). The animal experiments were conducted according to the guidelines approved by the Institutional Animal Care and Ethical Committee of the University of Shizuoka (Approval No. 176233).

#### 3.6.2. Measurement of Plasma Calcium Levels

To evaluate the pharmacological action of sCT samples, plasma calcium levels were measured after intratracheal administration (40 µg-sCT/kg, i.t.). For the intratracheal administration, DP-4 (Penn-Century, Philadelphia, PA, USA), a powder insufflator, was used to deliver the sCT samples accordingly. As a reference group, the plasma calcium level after intratracheal administration of sCT-RP (40 µg/kg, i.t.) was measured. After insufflation of sCT samples, blood (100 µL) was collected from the tail vein and centrifuged at 10,000× *g* for 10 min to obtain plasma samples. The plasma calcium levels were determined using a commercially available kit (Metallo Assay Calcium (CPZIII) kit, Metallogenics Co., Ltd., Chiba, Japan).

### 3.7. Statistical Analysis

The data were analyzed using a one-way analysis of variance with a Fisher’s least significant difference test. *p*-values of less than 0.05 were considered to show a significant difference in all analyses.

## 4. Conclusions

In the present study, the FDD process was applied to the particle design for pulmonary delivery of sCT, a model therapeutic peptide drug. The FDD process could successfully produce sCT-loaded PLGA microspheres for DPI with a uniform size distribution regardless of the low glass transition point of the polymeric carrier. In dissolution tests using SLF to mimic the conditions in the respiratory tract, the release behavior of sCT could be controlled by encapsulation into PLGA microspheres. The DPI system of sCT-SR extended the duration of the hypocalcemic action after intratracheal administration in rats, due to its fine inhalation performance and sustained-release property. From these observations, strategic application of the FDD process could be an efficacious option to prepare PLGA-based DPI of sCT, as well as other therapeutic peptides and proteins, to enhance their therapeutic potential.

## Figures and Tables

**Figure 1 molecules-25-01311-f001:**
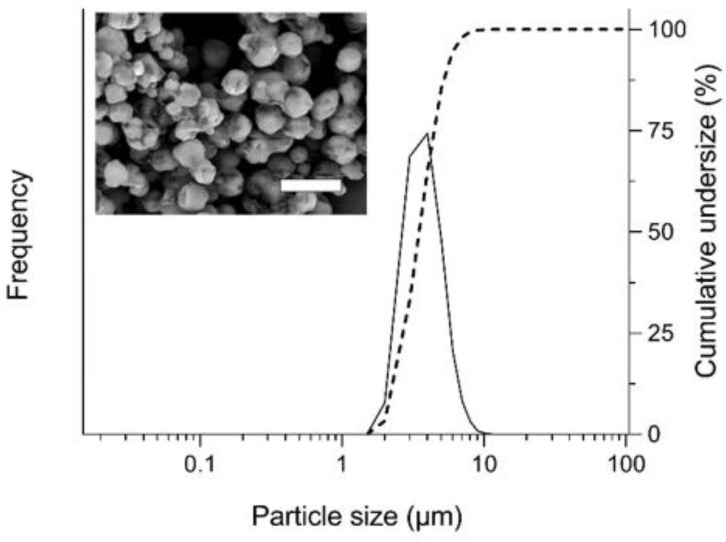
Appearance and particle size distribution of sustained-release particles of salmon calcitonin (sCT/SR) prepared with the fine droplet drying (FDD) process. Solid line, frequency; dotted line, cumulative undersize fraction curve. White bar represents 10 µm.

**Figure 2 molecules-25-01311-f002:**
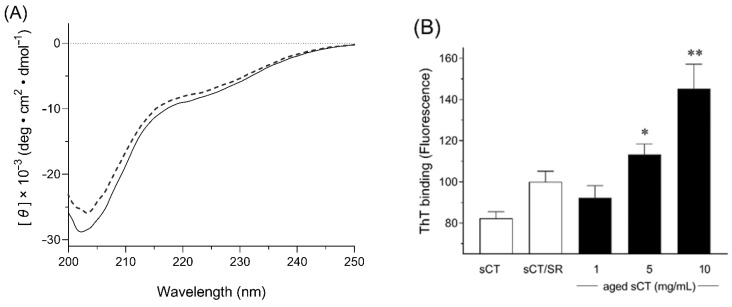
Changes in secondary structure of sCT during FDD process. (**A**) Circular dichroism (CD) spectrum of fresh sCT solution (solid line) and extracted sCT solution from sCT/SR (dotted line). (**B**) ThT binding assay of sCT samples. The aggregated sCT in diluted solutions were detected by the thioflavin T binding assay. Non-aged sCT, and sCT/SR were dissolved in 20 mM PBS (pH 7.4). *, *p* < 0.05; and **, *p* < 0.01 *vs.* non-aged sCT. Each bar represents mean ± SE of 3 independent experiments.

**Figure 3 molecules-25-01311-f003:**
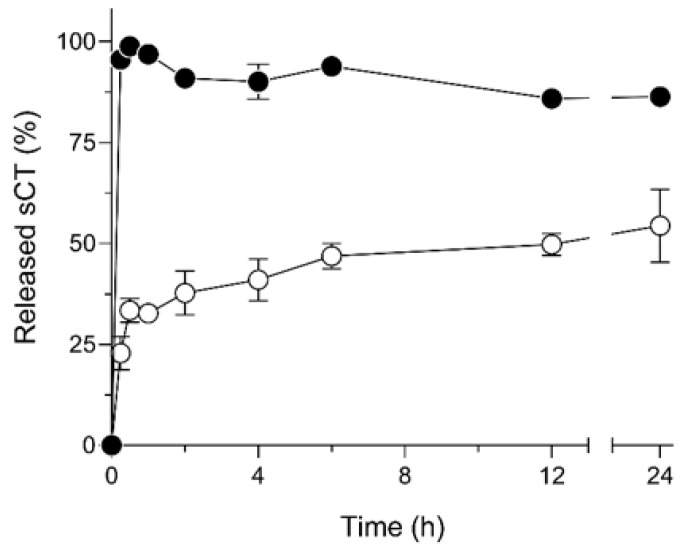
In vitro release behavior of sCT from PLGA microparticles in simulated lung fluid. ●, sCT powder; and ○, sCT/SR. Each bar represents mean ± SE of 3 independent experiments.

**Figure 4 molecules-25-01311-f004:**
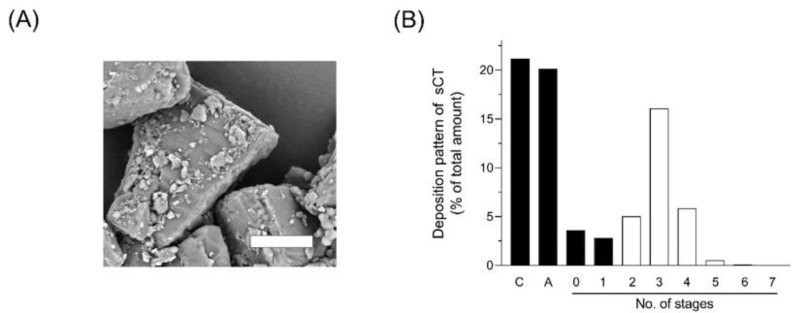
Appearance and in vitro inhalation property of sCT/SR-respirable powder (RP). (**A**) Scanning electron microscopic image of sCT/SR-RP. White bar represents 25 µm. (**B**) Deposition pattern of sCT/SR-RP in Andersen cascade impactor (C, capsule; and A, adapter). The analysis was conducted using a Jet haler^®^ with an airflow rate of 28.3 L/min. The fine particle fraction (FPF) value of sCT/SR-RP was calculated using the ratio of total drug deposited in stage 2 and lower (white bars).

**Figure 5 molecules-25-01311-f005:**
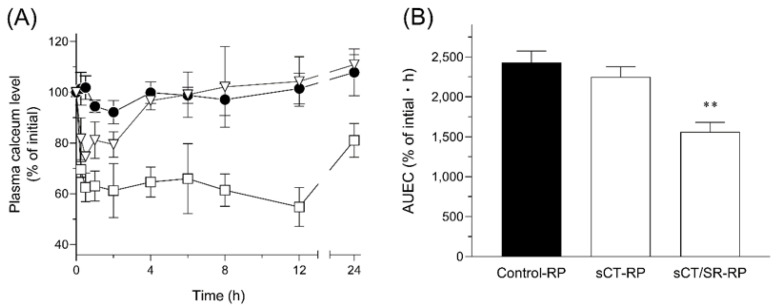
Hypocalcemic action of sCT after the intratracheal administration of sCT samples in rats (40 µg-sCT/kg, i.t.). (**A**) Time transitions of plasma calcium levels after the administration of sCT samples and (**B**) AUEC_0–24 h_ of each group. ●, control-RP; ▽, sCT-RP; and □, sCT/SR-RP. Data represent mean ± SE of 4–6 experiments. **, *p* < 0.01 with respect to the control-RP group.

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
