# Peer review of "Design and Characterizations of Inhalable Poly(lactic-co-glycolic acid) Microspheres Prepared by the Fine Droplet Drying Process for a Sustained Effect of Salmon Calcitonin"

_molecules, 2020, doi:10.3390/molecules25061311_

Round 1

Reviewer 1 Report

An interesting paper using inkjet printing to produce microspheres.  There are several other papers on microsphere preparation using printing technologies.  It will be good if authors of this paper could put a brief summary of the state of the art regarding the printing of microspheres in the introduction, before talking about FDD.

Line 41 – rephrase for correct English

Line 74 – DIPs?

Line 153, define SLF before giving the abbreviation

Line 158, ST or SR?

Line 108, define what you mean by ‘single micron particles’

Line 167 – define DPPC in full before giving abbreviation

Line 176, what is RP?

Line 183, how were microparticles of sCT/SR adhered on to the surface of lactose carrier?

Line 184, how do you know that ‘flow ability of RP formulation was better than sCT/SR?

Line 193, define FPF in full before giving abbreviation

In Figure 4 B, what do black and white bars represent?

Line 248, re 2% w/w solid contents.  How much was PLGA and how much was sCT?

Line 332, vein is misspelled

Author Response

We thank the peer reviewers for their thoughtful comments.  Our responses to the reviewers are as follows:

Reply to Reviewer#1

Comment 1: An interesting paper using inkjet printing to produce microspheres.  There are several other papers on microsphere preparation using printing technologies.  It will be good if authors of this paper could put a brief summary of the state of the art regarding the printing of microspheres in the introduction, before talking about FDD.

We thank for the invaluable comments of the reviewers.  The reviewer's suggestion would be helpful to improve the quality of our research.  We have revised the manuscript thoroughly as the reviewer pointed out.  We hope our responses are satisfactory one’s for you.  As per the suggestion, a brief summary of previous reports on ink jet printing has been added in the introduction section as follows; “Recently, several ink jet printing technologies have been applied to the production of pharmaceutical formulations in the form of particles, hydrogels, and films for different administration routes, including oral, transdermal, respiratory or parenteral routes [15-17], since the industrial inkjet printing has now reached high standards of flexible, robust and reliable performance.” (Page 2,Line 59)

  1. Scoutaris, N.; Ross, S.; Douroumis, D., Current Trends on Medical and Pharmaceutical Applications of Inkjet Printing Technology. Pharmaceutical research 2016, 33, (8), 1799-816.
  2. Daly, R.; Harrington, T. S.; Martin, G. D.; Hutchings, I. M., Inkjet printing for pharmaceutics - A review of research and manufacturing. International journal of pharmaceutics 2015, 494, (2), 554-567.
  3. Buanz, A. B. M.; Belaunde, C. C.; Soutari, N.; Tuleu, C.; Gul, M. O.; Gaisford, S., Ink-jet printing versus solvent casting to prepare oral films: Effect on mechanical properties and physical stability. International journal of pharmaceutics 2015, 494, (2), 611-618.

Comment 2: Line 41 – rephrase for correct English

              We appreciate the comment.  The phrase has been corrected as below; “For the administration of the biologics, parenteral routes like intravenous and/or intramuscular injections have been mainly used due to their poor oral bioavailability and stability in the gastrointestinal tract, which can adversely affect patient compliance.” (Page 1, Line 40)

Comment 3: Line 74 – DIPs?

Comment 4: Line 153, define SLF before giving the abbreviation

Comment 5: Line 158, ST or SR?

Comment 6: Line 167 – define DPPC in full before giving abbreviation

Comment 7: Line 176, what is RP?

Comment 8: Line 193, define FPF in full before giving abbreviation

              We are deeply sorry for typos and missing information of some abbreviations.  We have revised the manuscript accordingly to define and correct these abbreviations as follows; “…….properties of DPIs,……” (Page 2, Line 78); “…….the release test in simulated lung fluid (SLF) was …….” (Page 4, Line 157); “On the contrary, sCT/SR exhibited ……” (Page 5, Line 164); “…… like dipalmitoylphosphatidylcholine (DPPC) owing to ……” (Page 5, Line 172); “2.3. Inhalation properties of sCT/SR-respirable powder (sCT/SR-RP)” (Page 5, Line 180); “The fine particle fraction (FPF) value is ……” (Page 6, Line 200)

Comment 9: Line 108, define what you mean by ‘single micron particles’

              We apologize for the confusing description.  The words have been replaced with “micron-sized particles”. (Page 3, Line 112)

Comment 10: Line 183, how were microparticles of sCT/SR adhered on to the surface of lactose carrier?

              We thank for the comment.  Generally, van der Waals force and electrostatic interactions can be the driving force for adhering the microparticles on to the surface of carrier particles.  To clearly show the information, the manuscript has been revised and added the reference article as below; “The micron-sized particles can be adhered by van der Waals force and electrostatic interactions, and the shape and surface roughness of the particle are significant factors to determine the strength of adhesion force [41].” (Page 5, Line 187)

  1. Adi, H.; Traini, D.; Chan, H. K.; Young, P. M., The influence of drug morphology on aerosolisation efficiency of dry powder inhaler formulations. Journal of pharmaceutical sciences 2008, 97, (7), 2780-8.

Comment 11: Line 184, how do you know that ‘flow ability of RP formulation was better than sCT/SR?

              We appreciate the comment.  For the preparation of DPI formulations, carrier particles like lactose are commonly used as an excipient to avoid the aggregation of micronized drug particles and improve the flowability of DPI formulations.  Generally, larger particle has better flowability than fine particles due to the difference of active surface area.  Thus, the adhesion of micron-sized particles on to large particle (lactose carrier) can improve the flowability by increasing the apparent particle size of DPI formulations.  Although the flow ability of sCT formulations were visually checked in the experiment, the description seems to be not appropriate as the reviewer pointed out.  The manuscript has been revised as below; “To avoid such aggregation and improve flowability of sCT/SR, the carrier particles of lactose with a mean diameter of ca. 50 mm were mixed,……” (Page 5, Line 185); “In the SEM image of sCT/SR-RP (Fig 4A), the aggregation of sCT/SR was negligible.” (Page 5, Line 189)

Comment 12: In Figure 4 B, what do black and white bars represent?

              We thank for the comment.  For better understanding of the data, we have added the definition of the white bar in the caption of figure 4.  “The FPF value of sCT/SR-RP was calculated using the ratio of total drug deposited in stage 2 and lower (white bars).” (Caption of figure 4)

Comment 13: Line 248, re 2% w/w solid contents.  How much was PLGA and how much was sCT?

              We are sorry for missing the information.  The detail amount of PLGA and sCT have been added in the method section as follows; “Briefly, sCT (50 mg) and PLGA (Resomer® RG-502H) (4,950 mg) were dissolved in ethyl acetate (245 g) (2%w/w as solid contents).” (Page 7, Line 253)

Comment 14 :Line 332, vein is misspelled

              We really appreciate the comment.  This typo has been corrected as per the reviewer’s comment.  “…… from the tail vein ……” (Page 9, Line 339)

Reviewer 2 Report

The article entitled “Design and characterizations of inhalable poly(lactic3 co-glycolic acid) microspheres prepared by fine droplet drying process for sustained effect of salmon calcitonin” is an interesting technological paper in which the issue of fabrication of inhalable PLGA microparticles has been assessed. The work is thorough and fairly organized/written, and can be published provided a few minor changes are carried out.

Lines 40-42: the sentence is not clear. I guess the “[…] which can adversely affect deterrent to medication adherence […]” should be “[…] which can adversely affect patient compliance […]”. Please, specify.

Line 92. Equimolar PLGA is completely amorphous; therefore, the term “melt” is not very correct, since PLGA progressively undergoes glass transition (at approximately 40-45°C in the dry state) and is then expected to turn to tge fluid state.

I also suggest that percent loading efficiency of sCT should be reported.

The bibliography is outdated (only 10/37 of the references are from 2015 onwards) and must be updated.

Author Response

We thank the peer reviewers for their thoughtful comments.  Our responses to the reviewers are as follows:

Reply to Reviewer#2

Comment 1: The article entitled “Design and characterizations of inhalable poly(lactic3 co-glycolic acid) microspheres prepared by fine droplet drying process for sustained effect of salmon calcitonin” is an interesting technological paper in which the issue of fabrication of inhalable PLGA microparticles has been assessed. The work is thorough and fairly organized/written, and can be published provided a few minor changes are carried out.

              We are really grateful for the expert comments and excellent advice we have received.  The thoughtful comments have really helped us to strengthen and improve the theory of the manuscript.  We have revised the manuscript thoroughly as the reviewer pointed out.  We hope our responses are still satisfactory ones for you.

Comment 2: Lines 40-42: the sentence is not clear. I guess the “[…] which can adversely affect deterrent to medication adherence […]” should be “[…] which can adversely affect patient compliance […]”. Please, specify.

              We appreciate the comment.  As the reviewer kindly pointed out, the sentence has been revised as follows; “For the administration of the biologics, parenteral routes like intravenous and/or intramuscular injections have been mainly used due to their poor oral bioavailability and stability in the gastrointestinal tract, which can adversely affect patient compliance.” (Page 1, Line 40)

Comment 3: Line 92. Equimolar PLGA is completely amorphous; therefore, the term “melt” is not very correct, since PLGA progressively undergoes glass transition (at approximately 40-45°C in the dry state) and is then expected to turn to tge fluid state.

              We thank for the invaluable comment.  As per the reviewer’s comment, the term “melt ” has been replaced with appropriate words accordingly.  “…… because the glass transition point of PLGA is ……” (Page 3, Line 97); “…… the low glass transition point of the polymeric carrier.”(Page 9, Line 350)

Comment 4: I also suggest that percent loading efficiency of sCT should be reported.

              We completely agree with the suggestion.  The loading efficiency of sCT has been added in the results and discussion section. “The loading efficiency of sCT into PLGA microsphere was 99.4% as assayed by HPLC analysis, indicating almost no degradation of sCT during the FDD process.” (Page 3, Line 99)

Comment 5: The bibliography is outdated (only 10/37 of the references are from 2015 onwards) and must be updated.

              We completely agree with the comment.  The reference information has been updated accordingly (21/46 of the references are from 2015 onwards).